# Genome Analysis of Epsilon CrAss-like Phages

**DOI:** 10.3390/v16040513

**Published:** 2024-03-27

**Authors:** Igor V. Babkin, Artem Y. Tikunov, Ivan K. Baykov, Vera V. Morozova, Nina V. Tikunova

**Affiliations:** 1Federal State Public Scientific Institution «Institute of Chemical Biology and Fundamental Medicine», Siberian Branch of the Russian Academy of Sciences, 630090 Novosibirsk, Russia; arttik@ngs.ru (A.Y.T.); ivan_baykov@mail.ru (I.K.B.); morozova@niboch.nsc.ru (V.V.M.); 2Shared Research Facility “Siberian Circular Photon Source” (SRF “SKIF”) of Boreskov Institute of Catalysis SB RAS, 630090 Novosibirsk, Russia

**Keywords:** crAss-like phages, genome, virus taxonomy, diversity-generating retroelements, reverse transcriptase, repressors/anti-repressors

## Abstract

CrAss-like phages play an important role in maintaining ecological balance in the human intestinal microbiome. However, their genetic diversity and lifestyle are still insufficiently studied. In this study, a novel CrAssE-Sib phage genome belonging to the epsilon crAss-like phage genomes was found. Comparative analysis indicated that epsilon crAss-like phages are divided into two putative genera, which were proposed to be named *Epsilonunovirus* and *Epsilonduovirus*; CrAssE-Sib belongs to the former. The crAssE-Sib genome contains a diversity-generating retroelement (DGR) cassette with all essential elements, including the reverse transcriptase (RT) and receptor binding protein (RBP) genes. However, this RT contains the GxxxSP motif in its fourth domain instead of the usual GxxxSQ motif found in all known phage and bacterial DGRs. RBP encoded by CrAssE-Sib and other *Epsilonunoviruses* has an unusual structure, and no similar phage proteins were found. In addition, crAssE-Sib and other *Epsilonunoviruses* encode conserved prophage repressor and anti-repressors that could be involved in lysogenic-to-lytic cycle switches. Notably, DNA primase sequences of epsilon crAss-like phages are not included in the monophyletic group formed by the DNA primases of all other crAss-like phages. Therefore, epsilon crAss-like phage substantially differ from other crAss-like phages, indicating the need to classify these phages into a separate family.

## 1. Introduction

CrAssphages were first discovered when analyzing NGS sequencing data using cross-assembly software [1]. CrAss-like phages are double-stranded DNA viruses that infect bacteria from the phylum Bacteroidetes [2]. Previously, all crAssphages were assigned to the order Crassvirales and divided into four groups: Alpha, Beta, Gamma, and Delta [3]. The International Committee on Taxonomy of Viruses (ICTV) recently approved new family names for crAss-like phages: the *Intestiviridae*, *Steigviridae*, *Crevaviridae*, and *Suoliviridae* families (corresponding to the former Alpha, Beta, Gamma, and Delta groups, respectively). These four families currently include 10 subfamilies, 42 genera, and 73 species [4].

The first crAss-like phage was isolated from a culture of *Bacteroides intestinalis* and named p-crAssphage or ΦCrAss001 (now *Kehishuvirus primaries* from the *Steigviridae* family, former Beta crAss-like phage group) [5]. Later, another 30 crAssphages were identified, cultivated, and studied, which allowed them to be classified to 11 different species of the *Steigviridae* family. Most of them infect *Bacteroides intestinalis*; however, three species infect *Bacteroides cellulosilyticus*, and one is specific to *Bacteroides tethaiotaomicron* [6,7]. In 2021, the ΦCrAss002 phage belonging to the *Jahgtovirus secundus* species and the *Intestiviridae* family (former Alpha group of crAss-like phages) was isolated from *Bacteroides xylanisolvens* [8]. Apart from these species, no other crAss-like phages have been isolated from bacterial cultures. Many cultured crAss-like phages require long-term cultivation and do not lyse planktonic cultures or form plaques on a lawn of sensitive cells even at high titers [4,8]. Metagenomic studies led to the discovery of the Gamma and Delta [3] and then the Zeta and Epsilon groups of crAss-like phages [4]. However, due to the difficulties of their cultivation, only their genomes were studied.

Despite the difficulties in culturing crAss phages, the genomic study of crAssphages has shown their important place in the intestinal microbiota and revealed their wide prevalence along with great diversity [4,9,10,11]. It has been shown that certain groups of crAss-like phages dominate in developed countries, while other crAss-like phages are more common among the rural populations of developing countries. The prevalence of crAss-like phages is substantially higher in the urban populations than in rural populations [6,12,13]. This fact can be partly explained by the association of *Bacteroides* with the diet of urban populations [14]. Due to the high prevalence of crAssphages in the human intestine, it is currently proposed to monitor their levels in treated wastewater to detect contamination with human feces [15,16,17].

Recently, two new groups of crAss-like phages have been identified among the intestinal circularized metagenome-assembled genomes (MAGs) [18]. These groups that were named Zeta and Epsilon showed substantial genetic distance from other crAss-like phages assigned to the four previously identified families. The crAss-like phages of the Epsilon group do not encode DNA polymerases, unlike other members of the Crassvirales order. Most Crassvirales carry a B-family DNA polymerase, while some of the *Stegiviridae* and Zeta crAssphages code for an A-family DNA polymerase [18]. It has been proposed that the crAss-like phages of Zeta and Epsilon groups should be separated into new families [4,18]. However, ICTV currently has not assigned them to any families.

In this study, a new epsilon crAss-like phage was found in the human gut virome and named crAssE-Sib. This phage genome encodes several proteins, including a DNA primase, virion RNA polymerase, reverse transcriptase (RT), and receptor binding protein (RBD), all of which substantially differ from those found in other crAss-like phages. In addition, crAssE-Sib encodes a prophage repressor and anti-repressors, which are highly conserved and could be involved in lysogenic-to-lytic life cycle switches. Finally, a detailed comparative analysis of CrAssE-Sib and related phages indicated that the Epsilon crAss-like phages should be divided into two putative genera.

## 2. Materials and Methods

### 2.1. Phage Genome Sequencing

Isolation of viral DNA from a fecal sample was performed as described in detail previously [19]. Briefly, the sample was clarified by repeated centrifugation and treated with DNase I (Thermo Fisher Scientific, Waltham, MA, USA) and Proteinase K (Thermo Fisher Scientific, Waltham, MA, USA). Then, phenol-chloroform extraction and subsequent ethanol precipitation were used to purify DNA. The resulting DNA was diluted in 50 µL of TE buffer, and DNA concentration was measured using Qubit 4.0 (Thermo Fisher Scientific, Waltham, MA, USA). A virome shotgun library (a truseq equivalent) was constructed using the NEB Next Ultra DNA library prep kit (New England Biolabs, Ipswich, MA, USA). For sequencing, a MiSeq Benchtop Sequencer and a MiSeq Reagent Kit 2 × 250 v.2 (Illumina Inc., San Diego, CA, USA) were used. After de novo assembly of the genome using the SPAdes software V.3.15.2 [20], the resultant single contig was further analyzed.

This work was approved by the Local Ethics Committee of the Center for Personalized Medicine, Novosibirsk (protocol #2, 12 February 2019). The written consent of the healthy volunteer was obtained in accordance with the guidelines of the Helsinki Ethics Committee.

### 2.2. Phage Genome Analysis

The genome annotation and analyses were performed in accordance with the guidelines described previously [21,22]. The termini of the genome that was obtained in a “pseudo-circular” form were determined with the help of PhageTerm [23] (https://galaxy.pasteur.fr, access date 25 August 2023). Online RAST server v. 2.0 [24] was used for the identification of ORFs and possible tRNA genes within the genome. tRNAscan-SE V 2.0 [25] was also used to predict the presence of tRNA genes. Genome function annotation was carried out by InterProScan (www.ebi.ac.uk/interpro/search/-sequence/, access date 3 October 2023), HHpred (toolkit.tuebingen.mpg.de/tools/hhpred, access date 3 October 2023), HHblits (toolkit.tuebingen.mpg.de/tools/hhblits, access date 3 October 2023), and BLASTP (NCBI, Bethesda, MD, USA, access date 3 October 2023). Predictions of domain architecture and functional motifs were made using CD-BLAST (NCBI, Bethesda, MD, USA, access date 3 October 2023). An e-value cut-off of 0.01 was used. Predictions of phage lifestyle from genomic data were made using PhageLeads (https://phageleads.dk/, access date 10 October 2023), PhageAI (https://app.phage.ai/, access date 10 October 2023), and BACPHLIP (https://cpt.tamu.edu/galaxy-pub?tool_id=edu-.tamu.cpt.bacphlip, access date 10 October 2023). 

### 2.3. Comparative Phage Analysis

The identification of nucleotide sequences that are similar to the crAssE-Sib phage genome was performed using BLASTN. Comparative analysis of crAssE-Sib phage proteome was performed by ViPTree version 3.7 web server (https://www.genome.jp/viptree, accessed on 3 August 2023) with default parameters [26]. ViPTree analysis based on genome-wide sequence similarities was computed by tBLASTx. The intergenomic similarity was determined using the VIRIDIC tool (http://rhea.icbm.uni-oldenburg.de/VIRIDIC, access date 17 October 2023) [27]. The genera were predicted using thresholds of the intergenomic identity of 70% [22,28].

### 2.4. Analysis of Proteins

Related amino acid sequences were downloaded from GenBank using crAssE-Sib phage proteins as a query in BLAST search against the NCBI protein database. Protein sequences were aligned using the M-Coffee method by the T-Coffee program [29]. The levels of amino acid identity of proteins were calculated by the BioEdit 7.2.5 program [30]. Maximum likelihood phylogenetic trees were generated by the IQ-tree program v. 2.0.5 [31]; the best-fit substitution model according to ModelFinder [32] was used. Branch supports were assessed using 1000 ultrafast bootstrap replicates [33]. The resulting trees were midpoint rooted and visualized in FigTree v1.4.1. Analysis of the genes included in DGR was carried out by the myDGR program [34]. The search for phage integrase genes was performed using programs IntegronFinder 2.0 [35] and FASTER [36]. Sequence logos representations of amino acid and nucleic acid multiple sequence alignment were calculated by WebLogo version 2.8.2 [37].

### 2.5. Three-Dimensional Modeling of Protein Structures

A three-dimensional model of the receptor binding protein (RBP) of crAssE-Sib phage was modeled using Alphafold2 (https://colab.research.google.com/github/sokrypton/ColabFold/blob-/main/AlphaFold2.-ipynb, access date 7 October 2023) [38] with a high degree of confidence (pLDDT > 90 for the most part of the molecule). Structural comparisons with experimentally determined 3D structures was performed using the Dali server (http://ekhidna2.biocenter.helsinki.fi/dali/, access date 10 October 2023) [39] and also using UCSF Chimera, version 1.17 [40]. Ribbon and surface representations of structural models were prepared using UCSF Chimera.

## 3. Results

### 3.1. Analysis of the Phage Genome

A virome shotgun library was constructed from a stool sample from a healthy volunteer (Project: PRJNA1025976; BioSample: SAMN37731570; SRA: SRR26322353). After sequencing and de novo assembly, a 151,256 bp gapless contig was identified using the SPAdes genome assembler. The average coverage was 126, and it was a pseudo-circular MAG, suggesting that the assembled sequence could be a complete phage genome. The starting point of the genome was predicted using PhageTerm analysis with a probability value 1.5 × 10^−22^. The name crAssE-Sib was proposed for this phage. The complete genome sequence of crAssE-Sib phage was submitted into the GenBank database with the accession number OR575929.

The genome of the crAssE-Sib phage codes for 196 putative ORFs and six tRNAs (Figure 1 and Appendix A). Of the 196 ORFs, 74 code for proteins with predicted functions or domains, which were determined based on the similarity of their amino acid sequences and domain structure with known phage proteins. The remaining 122 ORFs were assigned as hypothetical proteins.

### 3.2. Comparative Analysis of the crAssE-Sib Phage Genome

Comparison of the crAssE-Sib genome using BLASTn revealed the highest sequence identity with members of a new group of Epsilon crAss-like phages. From this group, sequences with a length of more than 140,000 bp were taken for further analysis. Previously, sequences of epsilon crAss-like phages of this length have been characterized as pseudo-circular MAGs, indicating that these sequences were complete phage genomes [18]. The analysis of the proteomic tree calculated using ViPTree confirmed that the crAssE-Sib phage belongs to the Epsilon crAss-like group of phages. SG values were calculated according to Bhunchoth et al. [41] as normalized tBLASTx scores (Figure 2 and Appendix A). In addition, comparative ViPTree analysis indicated that Epsilon crAss-like phages formed two clades: clade I and clade II. Clade I contains 35 phage genome sequences, including crAssE-Sib. Clade II contains 30 phage genome sequences (Figure 2 and Appendix A). All Epsilon crAss-like phages have similar genome lengths ranging from ~144 to 159 kb. The other crAss-like phages infect members of the Bacteroidota with sixteen *Cellulophaga* phages and one *Polaribacter* phage. The lengths of their genomes are shorter than those of Epsilon crAss-like phages and vary from 42 kb to 149 kb; however, most of the genomes are 70–80 kb. Significant differences between Epsilon crAss-like phages proteomes from their close relatives and from other crAss-like phages allowed us to classify them into a separate family.

To clarify the taxonomy of Epsilon crAss-like phages, a matrix of intergenomic similarities was constructed using the Virus Intergenomic Distance Calculator (VIRIDIC). A comparative genome analysis was performed on the base of intergenomic similarity, aligned genome fraction, and genome length ratio. The results confirmed the division of epsilon crAss-like phages into two subgroups, which represent the two proposed genera, *Epsilonunovirus* and *Epsilonduovirus*, in the group of Epsilon crAss-like phages (Figure 3). The data show that there are at least 31 different species in the genus *Epsilonunovirus*, including crAssE-Sib, and 25 species in the genus *Epsilonduovirus*. In some cases, individual species are represented by 2–3 genomes.

A search for prophage sequences similar to crAssE-Sib in bacterial genomes identified similar sequences in the genomes of *Parabacteroides distasonis* BFG-238 and *Parabacteroides merdae* CL06T03C08. A matrix of intergenomic similarities indicated that the prophage sequence of *P. distasonis* BFG-238 might be a member of a third putative genus within the group of Epsilon crAss-like phages (Appendix A).

Comparative genome alignment of the Epsilon crAss-like phages obtained using the ViPTree tool indicated a high level of gene synteny both within the proposed genera and between them (Figure 4). To eliminate highly similar sequences, only sequences with minimal nucleotide similarity (from 0.582 to 0.700) were used for the analysis. A total of 20 *Epsilonunovirus* (including crAssE-Sib) sequences and 10 *Epsilonduovirus* sequences were identified (Figure 4). 

At the beginning of these crAss-like phage genomes, there is a region from about 17 to 35 kb (in the crAssE-Sib genome) that is highly homologous among the related genomes. This region includes the genes encoding the terminase subunits and genes of structural proteins, namely portal, major capsid, and ring proteins. In addition, a significant homology was found in some other proteins of phages from both the *Epsilonunovirus* and *Epsilonduovirus* genera (carbohydrate-binding protein, muzzle protein, and some proteins of the replication apparatus).

### 3.3. Analysis of the crAssE-Sib Proteins

To verify the taxonomy of the crAssE-Sib phage and relative phages, phylogeny of the large subunit of terminase and DNA primase along with their most similar orthologs was analyzed (Figure 5). 

On the maximum likelihood phylogenetic tree of the terminase large subunit (Figure 5), sequences of Epsilon crAss-like phages are divided into two clades according to the *Epsilonunovirus* and *Epsilonduovirus* genera, as shown above in Figure 2 and Figure 3. In addition, they form an outer group compared to the rest of the crAss-like phages, which form a separate clade. This fact indicates that Epsilon crAss-like phages should be isolated into a separate family. Notably, there are two additional sequences that are closely related but are not members of the proposed genera *Epsilonunovirus* and *Epsilonduovirus*. However, these sequences of the terminase large subunit are derived from short (less than 30 kb) nucleotide sequences, and therefore, taxonomic analysis of these phages is impossible. 

Phylogenetic analysis of the phage primase showed that, as in the case of the large terminase subunit, two clades corresponding to the genera *Epsilonunovirus* and *Epsilonduovirus* can be distinguished (Figure 6). However, the primase sequences of Epsilon crAss-like phages significantly differ from those of other crAss-like phages (Figure 6), and the cluster of Epsilon crAss-like sequences is not part of the monophyletic group formed by the primase sequences of all other crAss-like phages [3,4,18].

Several other genes that substantially differed from those in other crAss-like genomes were found in crAssE-Sib. One such gene encoding the RNA polymerase was explored further (Figure 1 and Appendix A). HHpred analysis showed that the virion RNA polymerase of crAssE-Sib is similar to that of the Cellulophaga phage phi14:2 (*Steigviridae*) [42] with 95% probability (coverage 46%, e-value 3 × 10^−3^). On the ViPTree dendrogram, the phage phi14:2 with a heterogeneous group of other phages formed a neighboring cluster with Epsilon crAss-like phages (Figure 2 and Appendix A). The crAssE-Sib virion RNA polymerase was conserved among Epsilon crAss-like phages and differed significantly from the RNA polymerases of other crAss-like phages (Figure 7).

An analysis of the crAssE-Sub genome revealed some genes encoding proteins that can potentially participate in the life cycle of this phage. Two transposase genes oriented in opposite directions were found (Figure 1 and Appendix A). Both transposases are members of the RNA-guided endonuclease TnpB protein family [43], but they have an aa identity level of only 18.2% when compared to each other (Appendix A). A study of the known Epsilon crAss-like phages sequences revealed that transposases genes are present only in some members of the *Epsilonunovirus* genus and are absent in other phages. The crAssE-Sib genome also encodes the NinG-like protein. This gene is present in the genomes of all Epsilon crAss-like phages and has been shown to be involved in Red recombination of phage Lambda [44]. In addition, genes encoding a transcriptional repressor, and two prophage anti-repressors were detected in the crAssE-Sib genome (Figure 1 and Appendix A) [45,46]. A repressor was found using HHblits with a probability of 99.17% and an e-value of 1.2 × 10^−13^, whereas BLASTP identified it with an e-value of 3 × 10^−15^. Two anti-repressors that are similar to known lysogeny to lytic switch anti-repressors found in other Podoviridae phage (Appendix A) were detected using HHblits with a probability of about 100% and an e-value of 3.7 × 10^−36^, while BLASTP found them with e-value of about 0. InterProScan and CD-BLAST detected two domains (BRO_N and KilAC) in both anti-repressors with high reliability. The alignment of these two prophage anti-repressor sequences indicated significant identity, which suggests that these anti-repressors are probably paralogs (Appendix A). The repressor and at least one of the anti-repressors were found in all members of the *Epsilonunovirus* genus, but not in the *Epsilonduovirus* genus. Thus, the genes of temperate phages were found with greater reliability in crAssE-Sub and other *Epsilonunovirus* genomes. 

Predictions of phage lifestyles from genomic data were made using BACPHLIP, PhageLeads, and PhagAI. Most programs such as BACPHLIP and PhageLeads perform predictions based on the search for genes specific to temperate and lytic phages; PhageAI is based on machine learning to classify phages into lytic or temperate based on their nucleotide sequences. It is known that these software tools sometimes make errors, especially in the case of phage genomes whose close relatives are unknown [47]. PhageLeads found no temperate phage genes. BACPHLIP detected only two genes encoding transposases and predicted that this phage was lytic with a probability of 98%. This failure of PhageLeads and BACPHLIP to annotate the crAssE-Sib genome may be because this genome substantially differs from those of well-studied phages. PhageAI classified crAssE-Sib phage as a temperate phage with a reliability of 85.8%. 

Even though PhageLeads and BACPHLIP did not identify crAssE-Sib as a temperate phage, or the level of predictions was low (PhageAI), the identification of the above genes allows us to assume the possibility of a temperate lifestyle of crAssE-Sib and at least several other phages from the *Epsilonunovirus* genus. The detection of repressor/anti-repressor genes is especially important because such genes have not been found in the genomes of lytic phages. Despite the presence of genes containing BRO_N and KilAC domains in the genomes of *Epsilonunovirus* phages, which are conserved in temperate phages, experimental confirmation of the life cycle of these phages is necessary.

### 3.4. Analysis of the Diversity-Generating Retroelement (DGR) in crAssE-Sib

DGR is a remarkable prokaryotic genetic system that uses reverse transcription to introduce mutations into specific regions (variable repeats) of target genes [48,49,50]. DGR cassettes are widely present in the genomes of various crAss-like phages, and some of these phages contain complete DGR cassettes, some contain only DGR fragments, and others contain none. DGR in the crAssE-Sib genome contains all the essential elements (Figure 8). The RT gene was found using HHpred analysis. The sequence located upstream to the RT gene was identified as the target gene that encodes RBP and contains the variable repeat (VR). The sequence that is responsible for initiation of the mutagenic homing (IMH) was found downstream to the VR. The template repeat (TR) with IMH* sequence was detected between the RBD and RT genes (Figure 8). All these sequences are essential elements of the complete DGR cassette [48,49,50]. 

Screening the available genomes of epsilon crAss-like phages showed that DGRs were found in several phages from the *Epsilonunovirus* genus, whereas such cassettes were absent in members of the *Epsilonduovirus* genus (Appendix A). All genomes from the genus *Epsilonunovirus* encoded conserved RBPs, and VRs in these RBPs showed a certain pattern despite the variability in their aa sequences (Figure 9). 

Three-dimensional (3D) modeling of RBP encoded by crAssE-Sib was performed using Alphafold2 (Figure 10). Comparison of the obtained model with protein structures from PDB using the Dali server showed that RBP has a structural similarity with proteins containing the C-type lectin domain (Figure 10 and Appendix A). The highest similarity of the crAssE-Sib RBP was revealed with the TaqVP protein (pdb 5VF4, DALI Z-score 19.7) that is encoded by the gene located in the DGR cassette found in *Thermus aquaticus* (Figure 10 and Appendix A) [51]. However, RBP of crAssE-Sib contained an additional N-terminal part that was absent in TaqVP. RBP of crAssE-Sib and also showed high similarity to the variable protein 1 from *Treponema denticola* (pdb 2Y3C, DALI Z-score 11.7), which is also hypermutated by DGR [52]. Notably, the structure of crAssE-Sib RBP differed significantly from that of the target Mtd protein, which was found in the DGR cassette of the BPP-1 phage (Figure 10) and is considered to mediate attachment of the BPP-1 phage to Bordetella receptors. In the Mtd protein, the VR region (shown in green) is located on a flat part of its surface, whereas it is hidden in a cavity in the case of the RBP of the crAssE-Sib phage. Given that the randomized VR region of the protein is typically involved in binding to the putative receptor, the RBP of a crAssE-Sib phage may only bind to a knob, ridge, or fiber-like part of its receptor.

Importantly, although several *Epsilonunovirus* phages encode full RT (Figure 8), the RT gene in other genomes was fragmented or deleted, indicating that retrohoming is not used by these members of the *Epsilonunovirus* genus. In addition, TRs were conserved in those DGR cassettes that were complete (Figure 9).

Among all crAss-like phages, complete or incomplete DGR cassettes have been previously found in members of *Intestiviridae* (Alpha) and *Suoliviridae* (Delta) genera and Zeta crAss-like phages [18]. Analysis of the RT phylogeny of the crAss-like phages indicated that sequences from the *Epsilonunovirus* genus formed a monophyletic group that substantially differed from those from other crAss-like phages (Figure 11).

Comparative analysis of the RT amino acid sequences was performed using HHpred and InterProScan in addition to CD BLAST. The resulting data indicated that a standard catalytic RT domain is present in the RT from crAssE-Sib. Like all reverse transcriptases, crAssE-Sib RT has two subdomains: fingers and palm with the catalytically active YxDD motif. The only difference was the presence of unusual GxxxSQ motif in the palm subdomain instead the GxxxSP motif specific to most bacterial and phage DGRs.

## 4. Discussion

In this study, the complete genome of a new phage, crAssE-Sib, belonging to a group of Epsilon crAss-like phages was found in the human virome. A detailed comparative analysis of the crAssE-Sib genome and genomes of available Epsilon crAss-like phages indicated that all Epsilon crAss-like sequences can be divided into two proposed genera: *Epsilonunovirus* and *Epsilonduovirus*. This result is confirmed both by analysis of the overall phage proteomes and phylogenetic analyses of phage proteins, including the large terminase subunit, primase, and RNA polymerase. Despite the similarity in the organization of the *Epsilonunovirus* and *Epsilonduovirus* genomes, the genes encoding the two different transposases, RT, prophage repressor, and anti-repressors were found only in some of the genomes of *Epsilonunoviruses*. 

In addition, the crAssE-Sib genome and some other *Epsilonunoviruses* contained several genes that differed significantly from their orthologs found in members of the *Intestiviridae*, *Steigviridae*, *Crevaviridae*, and *Suoliviridae* families and the Zeta group of crAss-like phages. It has been previously suggested that DNA primase is the only protein that is conserved for all crAss-like phages, with the exception of Epsilon ones. This protein forms a monophyletic clade that probably descended from one crAss-like phage ancestor [3,4,18]. In this study, the primase genes found in the genomes of Epsilon crAss-like phages differed and were not included in the monophyletic clade with primases of other crAss-like phages (Figure 6). We also found the gene encoding RNA polymerase in the crAssE-Sib genome (Figure 7). This protein has a similar structure to virion RNA polymerase of the phage phi14:2 [42].

Our results showed that the crAssE-Sib genome contains a DGR cassette with all the essential elements, namely the RT gene and target gene encoding RBP, VR, TR, IMH, and IMH* (Figure 8). It has been previously shown that the RT encoded by DGR plays an important role in mutagenic retrohoming, a process in which an RNA transcript from the TR is reverse-transcribed by the RT and then inserted instead of VR into the target gene [48,49,50]. Therefore, RBP could potentially undergo mutagenic retrohoming in the crAssE-Sib phage. Despite the fact that the RT gene was detected with high confidence in the crAssE-Sib phage genome, the myDGR program [34] did not find the presence of DGR in crAssE-Sib and other Epsilon crAss-like phage genomes. This result can be explained by the presence of the GxxxSP motif in the fourth domain of crAssE-Sib RT instead of the characteristic GxxxSQ motif found in DGRs. Usually, the GxxxSP motif is detected in most retroviral, retrotransposon RTs and group II intron maturases [53] and finding this motif in RT from the DGR cassette was unexpected. When we changed this motif in RT of crAssE-Sib in silico, myDGR perfectly detected the RT gene and hence the DGR. It should be noted that myDGR is widely used to search and study DGRs. The presence of RT with unusual aa motifs in DGRs can prevent the detection and study of such DGRs. 

As for the RBP of crAssE-Sib, the target protein in DGR, no similar phage proteins were found in the Protein Data Bank (PDB). However, despite low sequence similarity, this RBP showed structural similarity with C-lectin fold-containing proteins that are the predominant type of target proteins in DGR cassettes [52]. C-type lectins are general ligand-binding proteins, so the presence of such a domain in the RBP of crAssE-Sib supports its receptor-binding role. Regarding other putative structural proteins of crAssE-Sib, there are no reference structures in PDB for the majority of them. Even though a precise cryo-EM model of the ΦcrAss001 crAss-like phage was recently published [54], it was not possible to determine the role of many crAssE-Sib putative structural proteins due to significant differences in genome organization of Epsilon crAss-like phages and Beta crAss-like phages.

Surprisingly, genes encoding prophage repressor and anti-repressors were found in some *Epsilonunovirus* genomes. It has been previously shown that the anti-repression system is associated with the switching of the life cycle in temperate phages [45,46,55,56]. Analysis of other crAss-like phage genomes showed that these genes are absent in other crAss-like phages. Attempts were made to detect prophage sequences related to Epsilon crAss-like phages in the bacterial genomes. We found two incomplete prophage sequences in the chromosomes of *P. distasonis* BFG-238 and *P. merdae* CL06T03C08. Comparative analysis of their proteomes with crAssE-Sib (genus *Epsilonunovirus*) and OLQV010 (genus *Epsilonduovirus*) suggested that these prophage sequences belong to unknown genera of temperate *Parabacteroides* phages. This putative genus could be the third genus in the Epsilon crAss-like phages family. Notably, the prophage sequence from *P. distasonis* BFG-238 demonstrates close levels of similarity with the *Epsilonunovirus* and *Epsilonduovirus* genera (Appendix A).

Based on the sequence similarity of CRISPR spacers in microbial genomes and sequences of crAss-like phage genomes, it has been previously supposed that Epsilon crAss-like phages infect bacteria from the *Bacteroides*, *Parabacteroides,* and *Porphyromonas* genera (class Bacteroidia) [9,18]. Thus, it is not likely that the crAssE-Sib phage also infects bacteria from this class.

Our search for the integrase gene in the Epsilon crAss-like phages genomes using FASTER and IntegronFinder 2.0 failed. Gene encoding of the NinG-like protein [44] involved in recombination processes has been discovered in crAssE-Sib genome. However, it is impossible to say whether the Epsilon crAss-like phage genomes can be inserted into the bacterial host genome. It has been shown that some temperate phages do not integrate their genomes into the host genome but save them as plasmids, and the genes of such phage genomes are expressed along with bacterial genes [57,58,59]. Possibly, the lifestyle of Epsilon crAss-like phages is determined by this mechanism. When crAss-like phages from the *Intestiviridae*, *Steigviridae*, *Crevaviridae,* and *Suoliviridae* families were studied, it was suggested that these crAss-like phages are lytic and incapable of lysogeny. However, these phages could be temperate (pseudolysogenic) since they are capable of long-term persistence in the host culture; some of them do not even lyse planktonic cultures after long-term cultivation [4,8]. In this case, the probable mechanisms of pseudolysogeny can be the carrier state or chronic infection with hibernation inside infected cells [4,5,60,61]. As for members of the genus *Epsilonunovirus*, their genomes encode elements required for true lysogeny. Finding of prophages that are related to this genus could confirm this assumption.

## Figures and Tables

**Figure 1 viruses-16-00513-f001:**
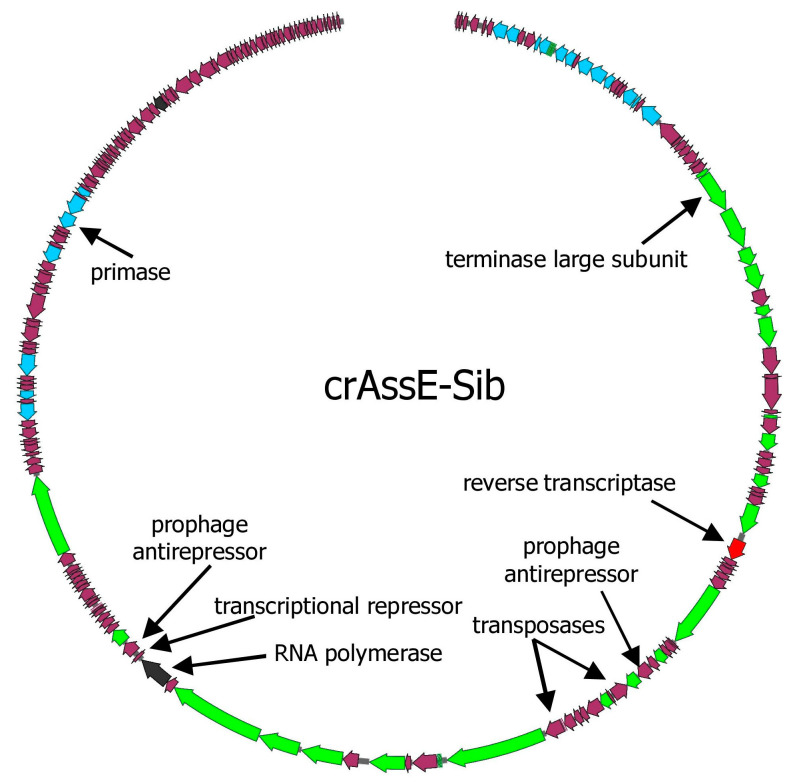
The whole genome map of crAssE-Sib phage. ORFs are colored according to their proposed function: DNA replication—blue; head assembly and structural—green; transcription—black; reverse transcriptase—red; other ORFs are colored with magenta.

**Figure 2 viruses-16-00513-f002:**
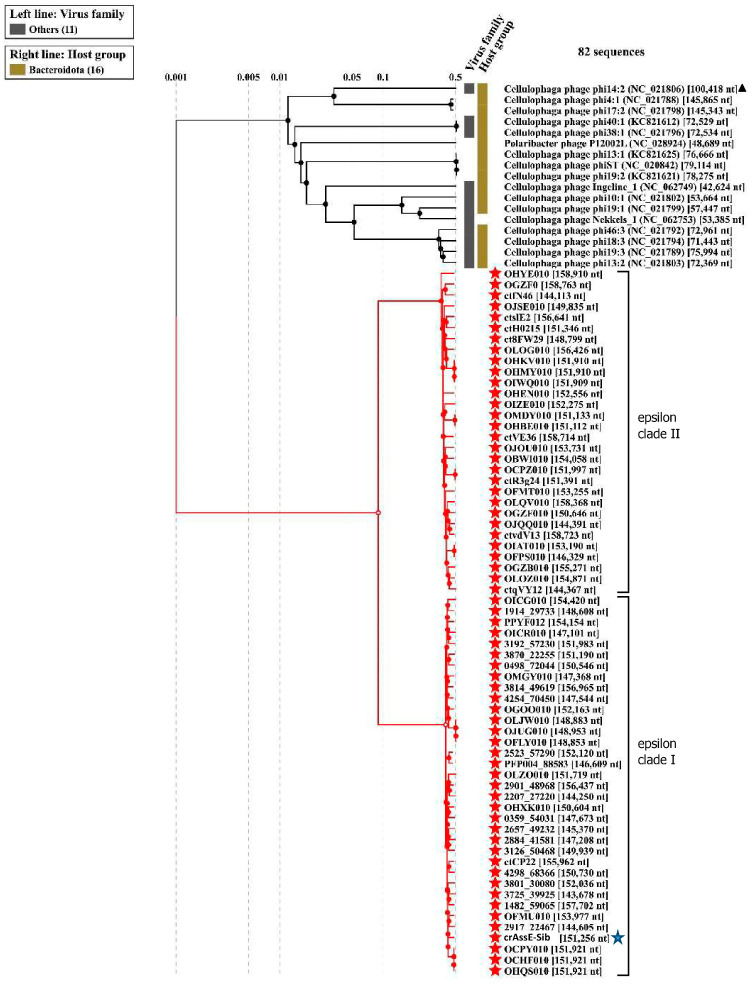
ViPTree analysis of the crAssE-Sib phage. Dendrogram plotted by ViPTree version 3.7 using crAssE-Sib and phages with SG values > 0.001. The crAssE-Sib phage is marked with a blue asterisk; phi14:2 phage is marked with a black triangle; phage sequences that were downloaded from the NCBI GenBank manually are marked with red phylogenetic branches.

**Figure 3 viruses-16-00513-f003:**
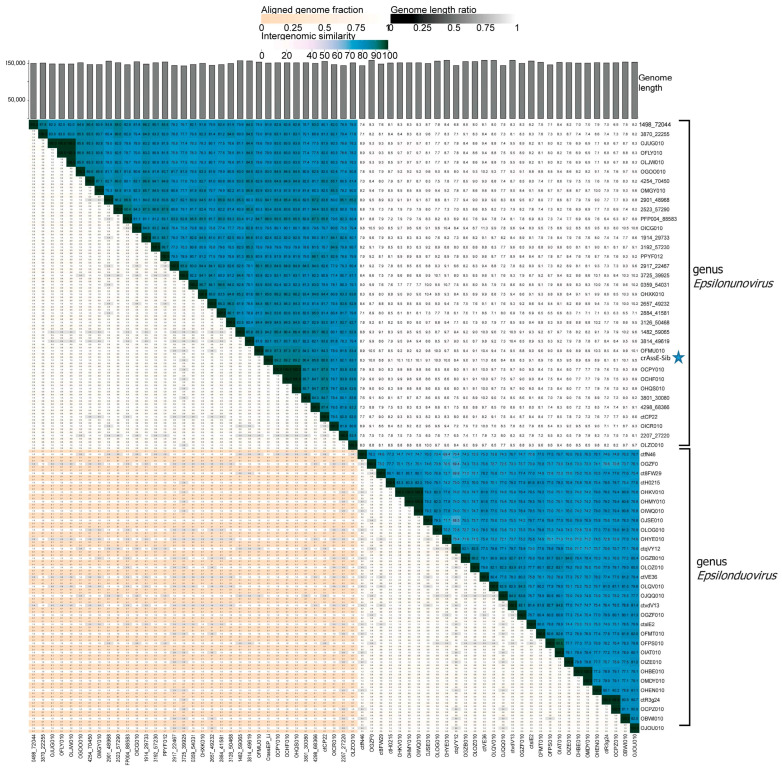
Bidirectional clustering heatmap visualizing VIRIDIC-generated similarity matrix for crAssE-Sib and related phage genomes. The crAssE-Sib phage is marked with a blue asterisk.

**Figure 4 viruses-16-00513-f004:**
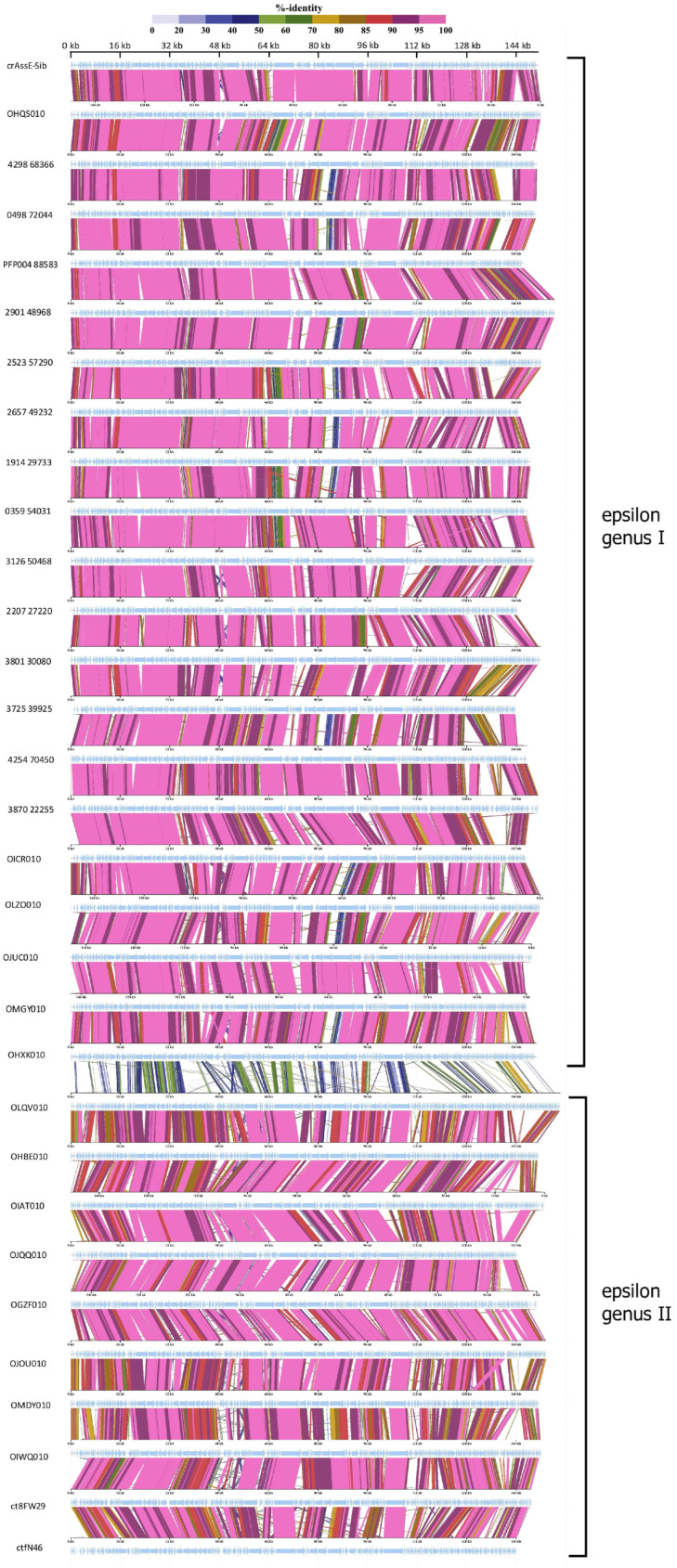
Comparative genome alignment of the crAssE-Sib and close phage genomes. Analysis was performed using VipTree software v. 3.7. The percentage of sequence similarity is indicated in color; the color scale is shown at the top.

**Figure 5 viruses-16-00513-f005:**
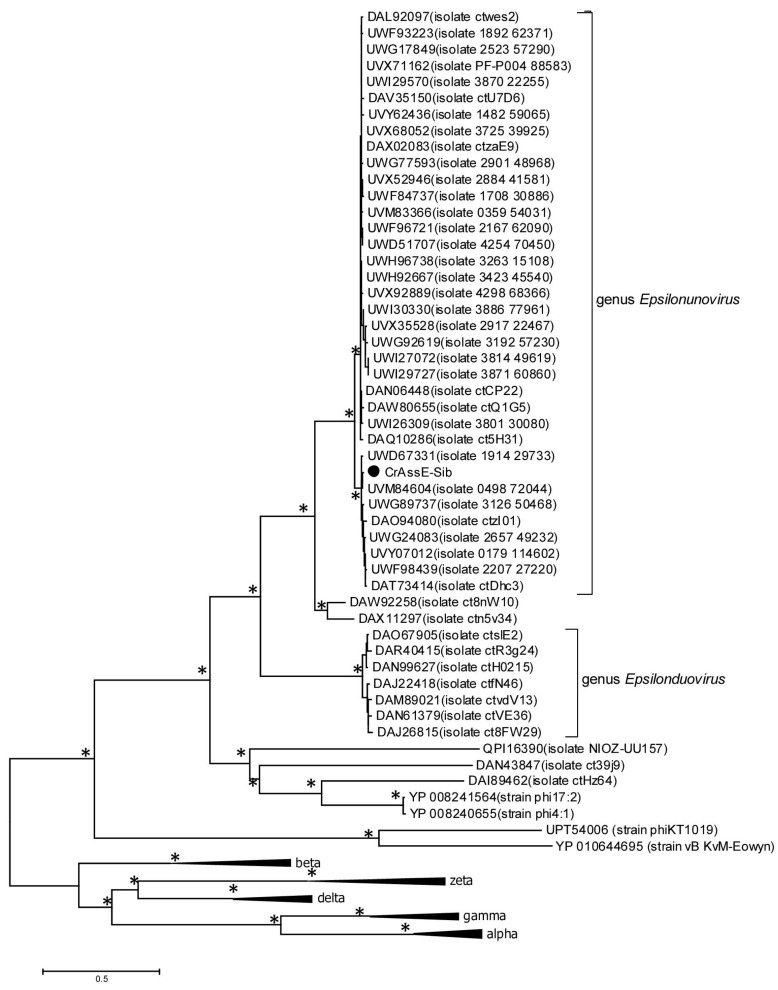
Maximum likelihood phylogenetic tree of the crAssE-Sib terminase large subunit generated using IQ-tree software v. 2.0.5. The crAssE-Sib phage is marked with a black circle. Nodes with 95% statistical significance calculated from 1000 ultrafast bootstrap (UFBOOT) replicates are marked with asterisks. The scale bar represents the number of substitutions per site.

**Figure 6 viruses-16-00513-f006:**
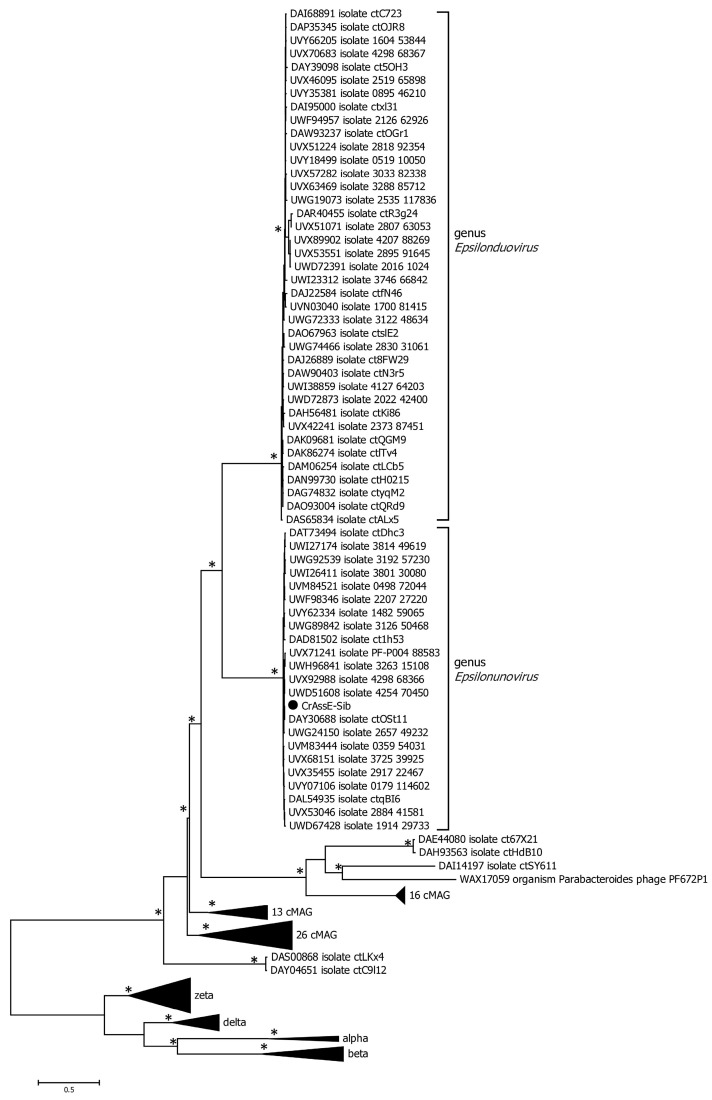
Maximum likelihood phylogenetic tree of the crAssE-Sib primase generated using IQ-tree software. The crAssE-Sib phage is marked with a black circle. Nodes with 95% statistical significance calculated from 1000 ultrafast bootstrap (UFBOOT) replicates are marked with asterisks. The scale bar represents the number of substitutions per site.

**Figure 7 viruses-16-00513-f007:**
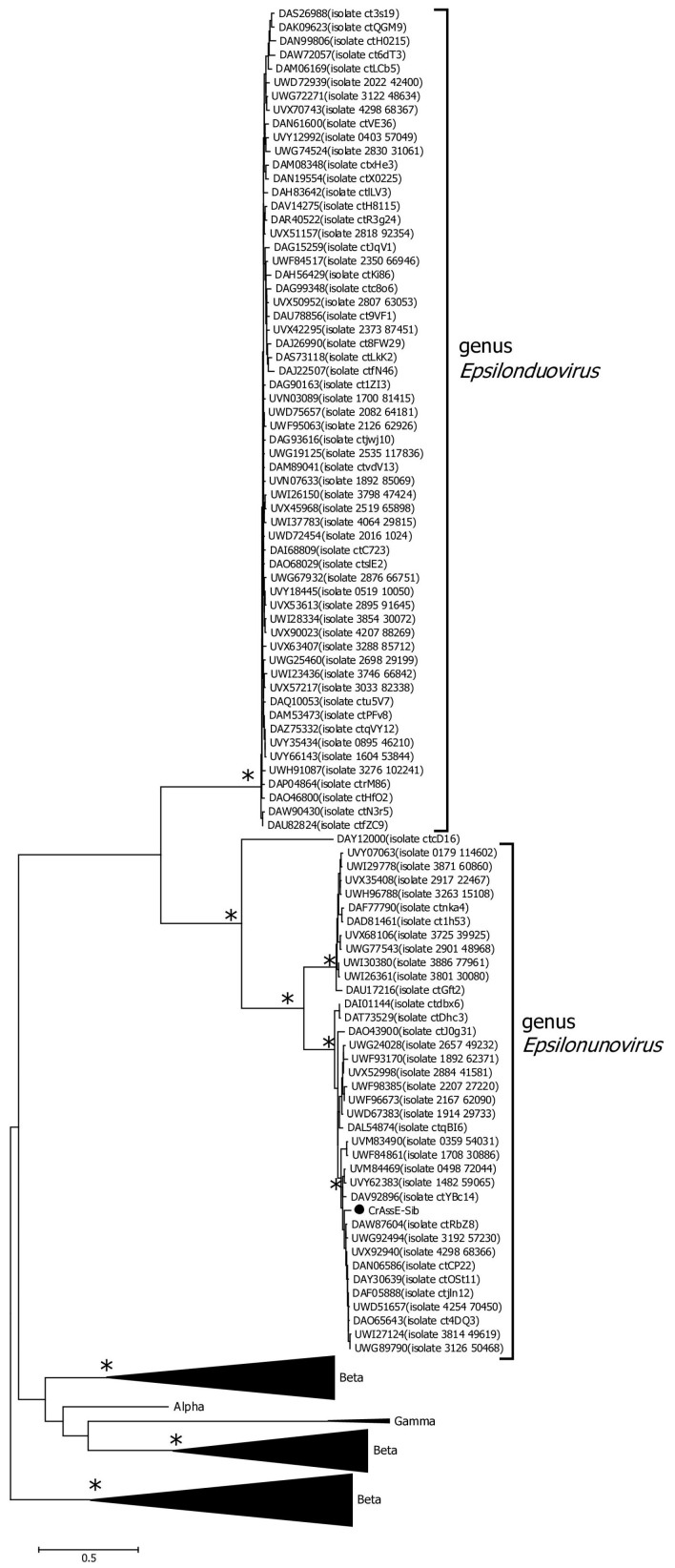
Maximum likelihood phylogenetic tree of the crAssE-Sib RNA polymerase generated using IQ-tree software. The crAssE-Sib phage is marked with a black circle. Nodes with 95% statistical significance calculated from 1000 ultrafast bootstrap (UFBOOT) replicates are marked with asterisks. The scale bar represents the number of substitutions per site.

**Figure 8 viruses-16-00513-f008:**
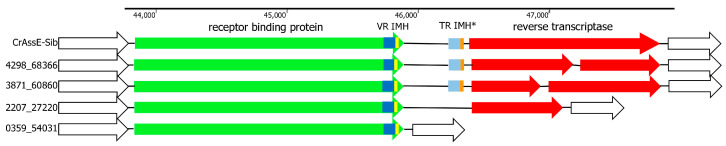
Schematic structure of DGR cassettes found in the genomes of crAssE-Sib and other members of the *Epsilonunovirus* genus. Schematic structure of DGR cassettes found in the genomes of crAssE-Sib and other members of the *Epsilonunovirus* genus. DGRs contain the receptor binding protein (RBP—green arrow) and reverse transcriptase (RT—red arrow) genes as well as template repeat (TR—light blue) with IMH* sequence (orange) and variable repeat (VR—blue) with initiation of the mutagenic homing (IMH—yellow) sequence. RT is fragmented or absent in many *Epsilonunoviruses*. Empty arrows indicate hypothetical proteins.

**Figure 9 viruses-16-00513-f009:**
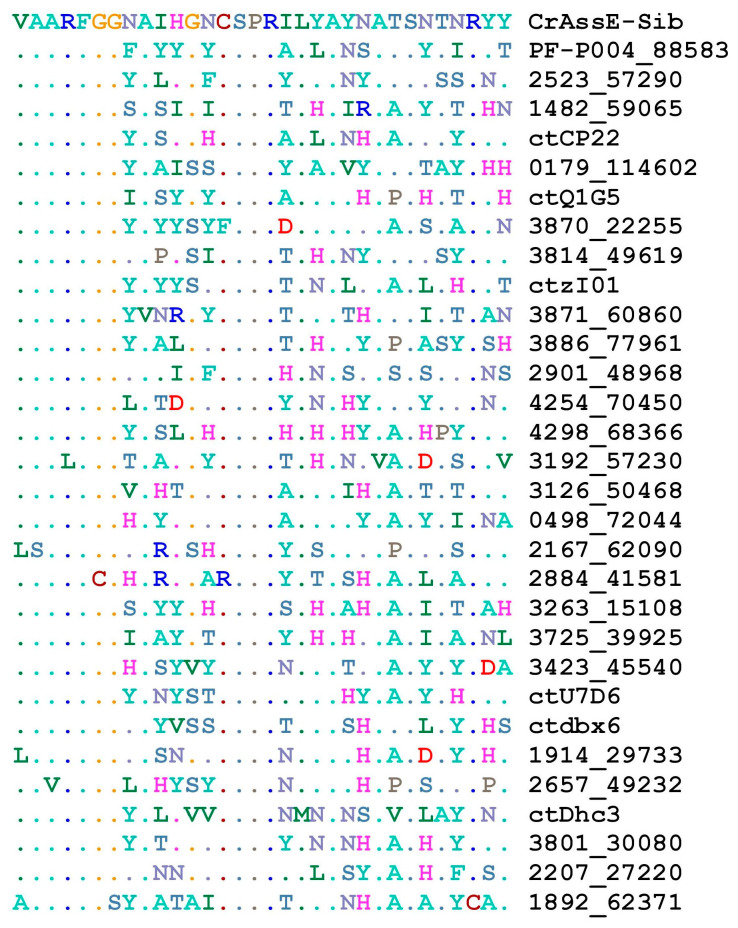
Multiple sequence alignment representations of aa sequences of the VRs region of crAssE-Sib and related phages.

**Figure 10 viruses-16-00513-f010:**
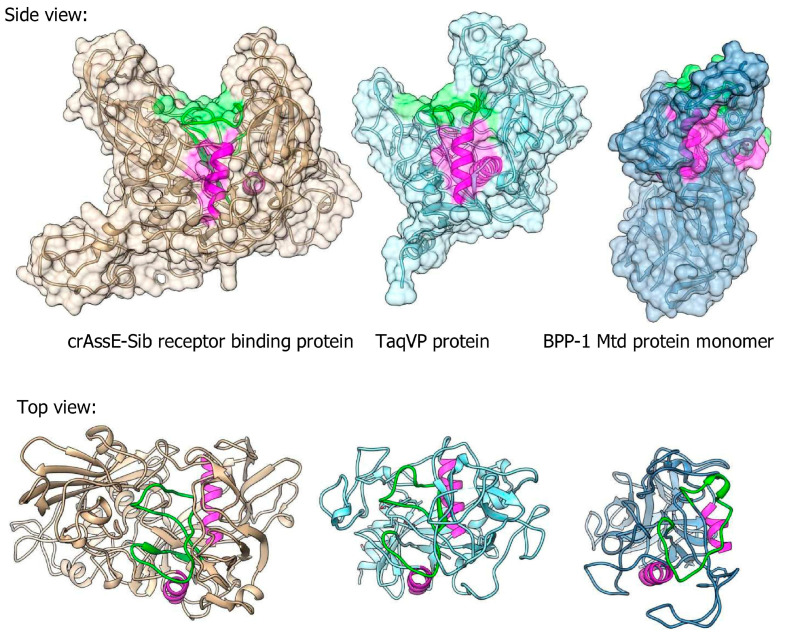
Comparison between 3D model of crAssE-Sib receptor binding protein (RBP) and experimental structures of TaqVP protein from *Thermus aquaticus* (pdb id 5VF4) and Mtd protein of the BPP-1 phage (pdb id 1YU0). VR regions are in green; magenta alpha helices show similar orientation of the molecules. All molecules are on the same scale.

**Figure 11 viruses-16-00513-f011:**
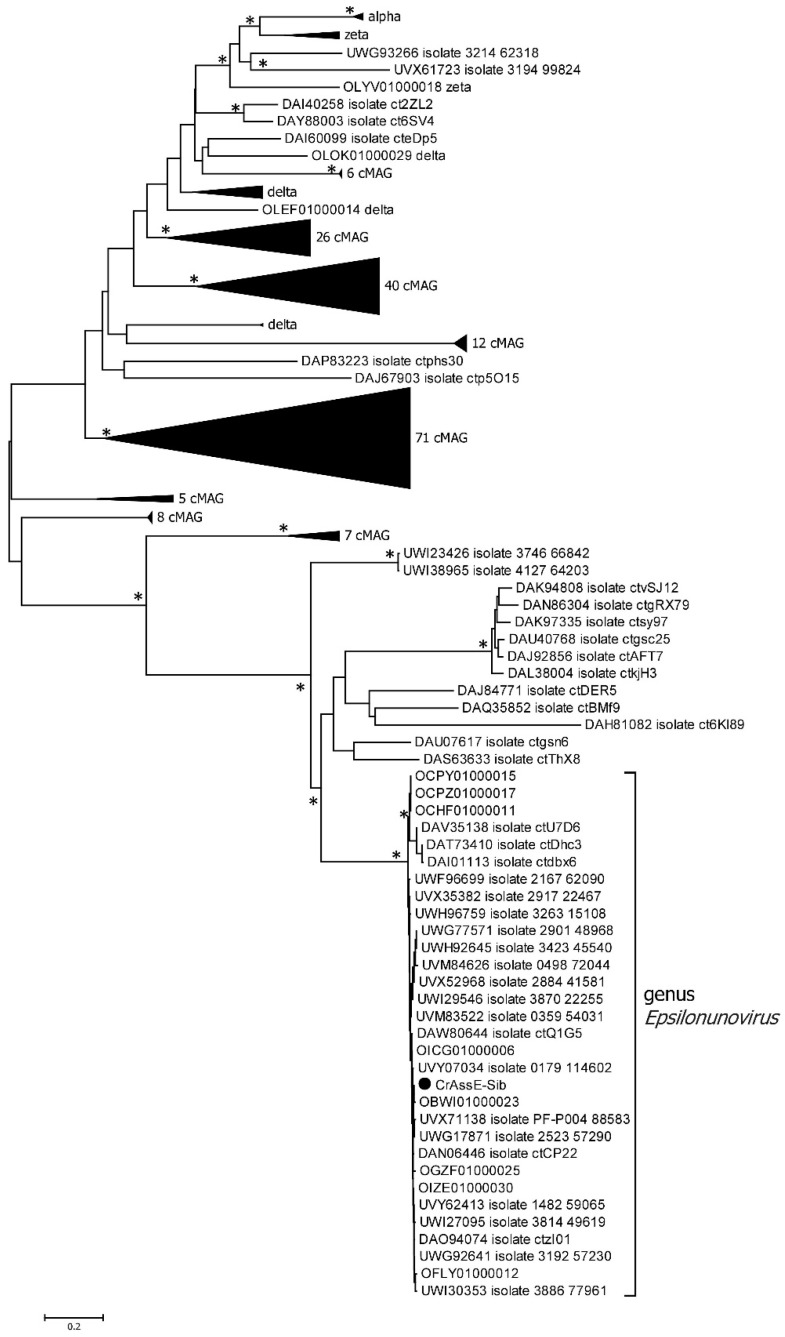
Maximum likelihood phylogenetic tree of the crAssE-Sib RT generated using IQ-tree software. The crAssE-Sib phage is marked with a black circle. Nodes with 95% statistical significance are marked with asterisks calculated from 1000 ultrafast bootstrap (UFBOOT) replicates. The scale bar represents the number of substitutions per site.

## Data Availability

The complete genome sequence of crAssE-Sib phage was submitted into the GenBank database with the accession number OR575929.

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
