# Peer review of "Genome Analysis of Epsilon CrAss-like Phages"

_viruses, 2024, doi:10.3390/v16040513_

Round 1

Reviewer 1 Report

Comments and Suggestions for Authors

This study reported a new type of phage called CrAssE-Sib that exists in the human gut virome. Based on comparative analyses of the phage's genomes, proteomes, and phylogenetics, the authors showed that epsilon crAss-like phages belong to a distinct family that differs from other crAss-like phages. The data is solid and the manuscript is well written. This study could expand our knowledge of epsilon crAss-like phages’ diversity, and also could contribute to the development of phage research. However, there are a few points that need attention to improve the manuscript's quality.

Line 35. When you use the abbreviation "ICTV" for the first time, please make sure to refer to its full name.

Line54-55. “The prevalence of crAss-like phages is substantially higher in the urban….”  Please clarify the sentence as it is confusing.

Line 281. “Notably, transposases genes were found ……” Please provide a reference or your data to support this.

Line 351. “N-terminal part that was absent in TaqVP…” To support your hypothesis. please show the sequence similarity between the RBP N-terminal domain and other proteins.

Figure 10. Please show the PBD code of TaqVP and Mtd in the figure.

Line 373-377. “The obtained data indicated that all necessary domains are present in…” Please provide the data to support this.

Author Response

We are grateful to the Reviewer 1 for the comments and made corrections and additions according all comments. Hope, this improved our manuscript.

This study reported a new type of phage called CrAssE-Sib that exists in the human gut virome. Based on comparative analyses of the phage's genomes, proteomes, and phylogenetics, the authors showed that epsilon crAss-like phages belong to a distinct family that differs from other crAss-like phages. The data is solid and the manuscript is well written. This study could expand our knowledge of epsilon crAss-like phages’ diversity, and also could contribute to the development of phage research. However, there are a few points that need attention to improve the manuscript's quality.

Line 35. When you use the abbreviation "ICTV" for the first time, please make sure to refer to its full name. 

Response: We added «International Committee on Taxonomy of Viruses»

Line54-55. “The prevalence of crAss-like phages is substantially higher in the urban….”  Please clarify the sentence as it is confusing.

Response: We have corrected the error

Line 281. “Notably, transposases genes were found ……” Please provide a reference or your data to support this.

Response: This is our data and we have noted it.

Line 351. “N-terminal part that was absent in TaqVP…” To support your hypothesis. please show the sequence similarity between the RBP N-terminal domain and other proteins.

Response: The hypothesis standing that N-terminal domain of RBP protein probably plays a role in the protein attachment to the virion is not supported by sufficient data, so we removed it.

Figure 10. Please show the PBD code of TaqVP and Mtd in the figure.

Response: We have changed the manuscript.

Line 373-377. “The obtained data indicated that all necessary domains are present in…” Please provide the data to support this.

Response: We have changed the manuscript

Reviewer 2 Report

Comments and Suggestions for Authors

This study was well carried out by clearly presenting comprehensive analysis of the crAssE-Sib phage genome. Analytical parts are well described along with bioinformatics tools, parameters, and algorithms. There are few comments.

1.      The genomic characteristics and taxonomic classification of the crAssE-Sib phage have been well presented, but need a little more interpretation regarding phage evolution, host interactions, and potential applications.

2.      Mostly computational tools for genome analysis and prediction have been used for descriptions, but need more statement regarding prediction validation.

3.      In conclusion, the evolutionary relationships of epsilon crAss-like phages seem to over hypothesized. It should be concluded based on the data obtained from this study.

Comments on the Quality of English Language

Moderate editing of English language required

Author Response

We thank the reviewer 2 for the comments on our work.

This study was well carried out by clearly presenting comprehensive analysis of the crAssE-Sib phage genome. Analytical parts are well described along with bioinformatics tools, parameters, and algorithms. There are few comments.

  1. The genomic characteristics and taxonomic classification of the crAssE-Sib phage have been well presented, but need a little more interpretation regarding phage evolution, host interactions, and potential applications.

Response: We made predictions in the article of the possible host of phages and their possible application to control the level of wastewater pollution by human faeces.

  1. Mostly computational tools for genome analysis and prediction have been used for descriptions, but need more statement regarding prediction validation.

Response: We have included in the manuscript the values of the reliability of predictions.

  1. In conclusion, the evolutionary relationships of epsilon crAss-like phages seem to over hypothesized. It should be concluded based on the data obtained from this study.

Response: As a result of our analysis, we can conclude that epsilon crAss-lake phages are evolutionarily significantly different from other crAss-like phages and at least two genera can be distinguished within them.

Moderate editing of English language required

Response: We have corrected the error

Reviewer 3 Report

Comments and Suggestions for Authors

The authors have done a commendable job on this manuscript titled "Genome Analysis of Epsilon Crass-like Phages Revealed That They Encode Diversity-Generating Retroelements with atypical Reverse Transcriptase" They present a very extensive analysis of a novel crAss phage crAssE-Sib in this report, starting with the annotated genome map of the phage. They present their findings of comparisions of crAssE-Sib with other crass phages at the nucleotide level using ViPTree and VIRIDIC, that suggest epsilon uno and duo viruses as generas of their own. They follow it up with phylogenetic analyses of crassE-Sib primase and RNA polymerase, as well the identification of repressor and anti-repressor genes through HHblits and BLAST. They hypothesize these proteins could be in involved in lysis-lysogeny switches and look into phage lifestyle predictions by multiple tools. By analyzing the sequences and structural predictions of different elements in the DGR element of crassE-Sib, like the RT and RBP, they show how epsilonunovriuses differ from others in these features. The authors conclude the article with a mostly well written discussion section, explaining and interpreting their findings.

While the work is scientifically quite sound, the overall writing style and language usage could be better. Plenty of suggestions have been made in the attached pdf with comments, that I would recommend the authors to go through and implement as they deem appropriate. Here are some other general comments:

1. I do not feel the current title captures the essence of the article in the right way. There is a lot more information in the paper other than the reverse transcriptase and also the core focus of the study crassE-Sib is missing from the title. I suggest rewriting a new title that mentions crassE-sib in it, and reflects the overall story conveyed through the paper. 

2. The interpretation of phage lifecycle predictions could be worded better. Additional information about tools could be added to help the user understand the difference between them and their results. 

3. Sometimes the fonts in the figures (both main manuscript and supplementary) look a little distorted. I would encourage the authors to work with the editorial team to ensure high quality images are used in the final version of the article for submission. Use fonts that are not affected by image size changes, and also make them visible enough to be readable.

4. Fig.9 needs extensive changes in terms of presentation of data (look in the attached pdf)

5. The direction column in Table S1, should be denoted as - or +, the usual notation rather than => and <=.

If these suggested changes are worked on, the manuscript could be reconsidered and a decision on acceptance can be made based on the revised version. I appreciate the authors attention to the details and efforts put into the various analyses and hope that the review helps in improving the presentation of their research. 

Comments on the Quality of English Language

Extensive comments on the quality of english language have been provided in the attached pdf. 

Author Response

We express our great gratitude to reviewer 3 for the proposed corrections of the manuscript. We accept all the proposed corrections and believe that they significantly improve the article.

The authors have done a commendable job on this manuscript titled "Genome Analysis of Epsilon Crass-like Phages Revealed That They Encode Diversity-Generating Retroelements with atypical Reverse Transcriptase" They present a very extensive analysis of a novel crAss phage crAssE-Sib in this report, starting with the annotated genome map of the phage. They present their findings of comparisions of crAssE-Sib with other crass phages at the nucleotide level using ViPTree and VIRIDIC, that suggest epsilon uno and duo viruses as generas of their own. They follow it up with phylogenetic analyses of crassE-Sib primase and RNA polymerase, as well the identification of repressor and anti-repressor genes through HHblits and BLAST. They hypothesize these proteins could be in involved in lysis-lysogeny switches and look into phage lifestyle predictions by multiple tools. By analyzing the sequences and structural predictions of different elements in the DGR element of crassE-Sib, like the RT and RBP, they show how epsilonunovriuses differ from others in these features. The authors conclude the article with a mostly well written discussion section, explaining and interpreting their findings.

While the work is scientifically quite sound, the overall writing style and language usage could be better. Plenty of suggestions have been made in the attached pdf with comments, that I would recommend the authors to go through and implement as they deem appropriate. Here are some other general comments:

  1. I do not feel the current title captures the essence of the article in the right way. There is a lot more information in the paper other than the reverse transcriptase and also the core focus of the study crassE-Sib is missing from the title. I suggest rewriting a new title that mentions crassE-sib in it, and reflects the overall story conveyed through the paper. 

Response: We have changed the title

  1. The interpretation of phage lifecycle predictions could be worded better. Additional information about tools could be added to help the user understand the difference between them and their results.

Response: We have corrected the article

  1. Sometimes the fonts in the figures (both main manuscript and supplementary) look a little distorted. I would encourage the authors to work with the editorial team to ensure high quality images are used in the final version of the article for submission. Use fonts that are not affected by image size changes, and also make them visible enough to be readable.

Response: We have corrected the figures. We have submitted figures with high resolution (600 dpi), which allows to consider even very small details. However, when processed on the mdpi site, they were overly compressed.

  1. Fig.9 needs extensive changes in terms of presentation of data (look in the attached pdf)

Response: We have corrected the figure 9.

  1. The direction column in Table S1, should be denoted as - or +, the usual notation rather than => and <=.

Response: We have corrected the Table S1.

If these suggested changes are worked on, the manuscript could be reconsidered and a decision on acceptance can be made based on the revised version. I appreciate the authors attention to the details and efforts put into the various analyses and hope that the review helps in improving the presentation of their research. 

Extensive comments on the quality of english language have been provided in the attached pdf. 

Response: We accept all the proposed corrections
